# Effect of Wastewater Irrigation on Photosynthesis, Growth, and Anatomical Features of Two Wheat Cultivars (*Triticum aestivum* L.)

**Shokoofeh Hajihashemi** [1,*], **Sonia Mbarki** [2,3], **Milan Skalicky** [2], **Fariba Noedoost** [1], **Marzieh Raeisi** [4] **and Marian Brestic** [2,5,*]

[1] Plant Biology Department, Faculty of Science, Behbahan Khatam Alanbia University of Technology, Khuzestan 63616-47189, Iran; nodost@bkatu.ac.ir

[2] Department of Botany and Plant Physiology, Faculty of Agrobiology, Food and Natural Resources, Czech University of Life Sciences, 16500 Prague, Czech Republic; mbarki@af.czu.cz (S.M.); skalicky@af.czu.cz (M.S.)

[3] Laboratory of Valorisation of Unconventional Waters. National Institute of Research in Rural Engineering, Water and Forests (INRGREF); BP 10, Ariana 2080, Tunisia

[4] Department of Environment, Faculty of Environment and Natural Resources, Behbahan Khatam Alanbia University of Technology, Khuzestan 63616-47189, Iran; marzieh_raeisi@gmail.com

[5] Department of Plant Physiology, Slovak University of Agriculture, A. Hlinku 2, 94976 Nitra, Slovakia

[*] Correspondence: hajihashemi@bkatu.ac.ir (S.H.); marian.brestic@uniag.sk (M.B.);
Tel.: +98-61-52721230 (S.H.); Fax: +98-61-52734781 (S.H.)

**Abstract:** The wastewater from the Razi petrochemical complex contains high levels of salts and heavy metals. In the present research, the effects of different wastewater dilution levels (0, 25%, 50%, and 100%) were studied on two wheat cultivars—Chamran and Behrang. The wastewater contained high levels of $NH_4^+$, $NO_3^-$, $PO_4^{3-}$, and $SO_4^{2-}$, and Mg, Ca, K, Na, Cu, Zn, Fe, M, and Ni. The toxic levels of mineral elements in the wastewater resulted in a significant decline in the K, P, Si, and Zn content of leaves. Irrigation with the wastewater resulted in a significant reduction in photosynthetic characteristics including chlorophyll fluorescence (Fv/Fm and $PI_{ABS}$), intercellular $CO_2$, net photosynthesis, water use efficiency, and photosynthetic pigments. The reduction in photosynthesis was followed by a significant decrease in the carbohydrate content and, subsequently, plant height, leaf area, and grain yield. Increasing the wastewater concentration reduced leaf thickness and root diameter, accounting for the decrease in xylem and phloem vessels, the root cortical parenchyma, and mesophyll thickness. The bulliform cell size increased under wastewater treatment, which may suggest induction of a defense system against water loss through leaf rolling. Based on the observed negative effect of wastewater on physiology, morphology, anatomy, and yield of two wheat cultivars, reusing wastewater with high levels of total suspended solids and salts for irrigation cannot be approved for wheat crops.

**Keywords:** growth; leaf anatomy; photosynthetic process; root anatomy; wastewater; wheat

## 1. Introduction

With the global shortage of water, major attention has been paid to reusing wastewater for irrigation of various crops. The wastewater irrigation of plants is a critical social and biological issue. Untreated or poorly treated wastewater can have harmful effects on soil, plants, and the environment, and be dangerous to humans. Depending upon the source of wastewater, it may contain high concentrations of salts and heavy metals. The high concentrations of salts may adversely affect physiological processes and growth of wastewater-irrigated plants, especially salt-sensitive

crops [1]. The Razi petrochemical complex is an important economic industry and a producer of chemicals such as ammonia, diammonium phosphate, phosphoric acid, and sulfuric acid [2,3]. The petrochemical operations produce a vast quantity of wastewater contaminated with different salts and heavy metals [2,4]. The wastewater produced by the petrochemical industry was found to be contaminated with a variety of toxic pollutants such as heavy metals [5–7] and using it for crop irrigation led to lower plant biomass and yield [1,8,9]. The heavy metal pollution interferes with the uptake of K, Mg, S, P, Ca, Fe, Mn, Zn and Cu and affects their translocation from roots to tops, which reduces the elements essential for plant growth [10–13]. There are some mineral elements which are critical for growth of crop plants and reduction of their concentrations from optimum levels impairs plant health and lowers yield [14].

It has also been reported that heavy metals inhibit the photosynthetic process [15]. The central atom of chlorophyll, magnesium, can be replaced by heavy metals like cadmium, copper, zinc, lead, and mercury, which blocks the light-harvesting capability of chlorophylls and impairs photosynthesis in stressed plants [15]. Photosynthesis involves the absorption of light energy and its conversion into biochemical energy. The chlorophylls absorb photons of light and measuring chlorophyll fluorescence is a useful tool for measuring photosynthetic energy conversion. An inverse correlation exists between photosynthetic efficiency and chlorophyll fluorescence, and there is a competition between the emission of fluorescent light and yield of photochemical energy [16–19]. Measurement of chlorophyll-a (chl-a) fluorescence has been introduced as a rapid tool to investigate the adverse effects of stress at an early stage of plant exposure to unfavorable growth condition [16,17,20–23]. Chl-a fluorescence can be used to estimate photosynthetic performance and consequently plant growth [24]. Decreased plant growth can be induced by disruption of photosynthesis or mineral nutrient imbalance, which causes a decline in yield [14,25].

The Razi petrochemical complex located in Mahshahr, Khuzestan province, is one of the most economically powerful companies in Iran [2,4]. Malmasi et al. (2010) reported that wastewater from this complex contained heavy metals. The reuse of industrial wastewater for agricultural purposes has increased because of water scarcity, but it is recommended to reuse reclaimed water and not wastewater directly. Furthermore, to avoid contamination of crops, soil, and public health issues, the treated wastewater for agricultural irrigation needs to meet minimum standards for specific pollutants. Some farmers believe that dilution of wastewater is a form of treatment that reduces its toxicity. Accordingly, this study was designed to measure the negative effects of different dilutions of untreated wastewater for irrigation of wheat as one of the most important crops worldwide. Because of the accumulation of heavy metals and salts in petrochemical industry wastewater, it is mandatory to determine the amounts of minerals and heavy metals in wastewater and to evaluate the effects of wastewater irrigation on wheat (*Triticum aestivum* L.). For this purpose, the effects of wastewater from the Razi petrochemical complex was studied on the two wheat cultivars, Chamran and Behrang, which are widely cultivated in this area. The response of wheat plants to water pollution was measured through evaluation of chlorophyll fluorescence, photosynthetic parameters and pigments, plant morphology and anatomy, mineral content in leaves, and grain yield.

## 2. Materials and Methods

### 2.1. Wastewater Sampling and Analysis

The wastewater was collected from the discharge channel of the Razi petrochemical complex, Mahshahr, Khuzestan, Iran. An inductively coupled plasma-optical emission spectrometer (Varian, 735 ICP-OES, Australia) was used to measure the elements in the wastewater, which contained various salts and heavy metals (Table 1). The wastewater was diluted with distilled water to prepare different concentrations: 0 (control), 25%, 50%, and 100% (undiluted) for irrigation of wheat plants.

**Table 1.** Elemental analysis of wastewater from Razi petrochemical complex.

| Parameter | Values (mg/L) | Parameter | Values (mg/L) |
|-----------|---------------|-----------|---------------|
| TSS | 325 | Na | 29,655 |
| Ag | 0.217 | Ni | 1.57 |
| Al | 3.2 | P | 99.5 |
| Ca | 1732 | Pb | 2.42 |
| Cd | 0.015 | S | 1702 |
| Co | 0.13 | Si | 12.06 |
| Cu | 0.892 | Zn | 8.07 |
| Fe | 2.98 | $NH_4^+$ | 346 |
| K | 1015 | $NO_3^-$ | 219 |
| Mg | 1374 | $PO_4^{3-}$ | 127 |
| Mn | 0.95 | $SO_4^{2-}$ | 680 |

All measurements were done according to standard methods for the examination of water and wastewater.

## 2.2. Plant Culture and Treatment

Seeds of the wheat (*Triticum aestivum* L.) cultivars, Chamran and Behrang, were obtained from Agricultural Jahad Behbahan, Khuzestan, Iran. The seeds were sterilized in ethanol (70% v/v) for 1 min, followed by incubation in sodium hypochlorite (20% v/v) for 20 min, and washing three times with sterile distilled water. The seeds were transferred to 40 Petri dishes prepared for each wheat cultivar; 10 seeds were placed in each dish. Dishes were transferred to a growth chamber at a controlled temperature of 25 ± 1°C and a photoperiod of 16 h of light and 8 h of dark for 14 days. Two weeks later, the seedlings with two leaves were transferred to 2 L pots filled with equal amounts of perlite and soil. Ten rooted plantlets with two leaves were planted in each pot. For each cultivar, the experiment was designed with eight pots for each treatment. The pots were placed in a phytotron with an air temperature of 25 ± 1 °C during the day (16 h) and 18 ± 1 °C at night (8 h), and relative humidity of 60 ± 2% during the experiment. The plants were irrigated every five days with four different concentrations of wastewater: 0 (distilled water), 25%, 50%, and 100% (undiluted). Ten days later, the seedlings were thinned to six similar plants per pot. The plants were harvested at two different stages of flowering (four months after sowing) and late seed maturation (six months after sowing). The culture and treatment experiments were repeated three times. Each experiment included eight pots for each treatment, with six plants per pot to provide enough plant material for physiological, morphological, and anatomical studies.

## 2.3. Chlorophyll Fluorescence and Photosynthesis Measurements

The chlorophyll fluorescence was measured with a portable chlorophyll fluorimeter (Pocket PEA, Hansatech, UK) on apical fully expanded leaves of 10 plants from different pots, at the 90th day after sowing. The maximum quantum yield of photosystem II ($F_v/F_m$) and the performance index of both photosystems I and II ($PI_{ABS}$) were measured early in the morning before the lights of the phytotron came on. The following day, photosynthetic characteristics including intercellular $CO_2$ ($C_i$), net photosynthesis ($P_N$), water use efficiency (WUE), and transpiration rate ($T_r$) were measured with a portable plant photosynthesis meter (KR8700 system; Korea Tech Inc., Seoul, Korea) [26]. The fully expanded apical leaves were harvested to measure photosynthetic pigments. The chlorophylls and carotenoids were extracted with acetone (80% v/v) and measured by spectrophotometer according to the Wellburn [27] assay.

## 2.4. Growth Analysis

The heights of 10 plants from different pots were measured at the 90th day after sowing. The area of the sixth fully expanded apical leaves in 10 different plants was measured using a leaf area meter (KR9700 system, Korea Tech Inc., Korea). The number of spikelets obtained from 10 plants was

recorded about six months after sowing the seeds. Five pots were harvested at the late seed maturation stage and the weight of grains was measured in every treatment.

## 2.5. Carbohydrate Measurements

Water-soluble carbohydrates (WSC) in the dry mass of sixth apical leaves were determined according to the DuBois et al. (1956) method. The glucose content of the same leaves was measured with a glucose assay kit (Sigma) [26].

## 2.6. Elemental Analysis of Seeds

The identification and quantitation of the elements in seeds was done using an inductively coupled plasma-optical emission spectrometer (Varian, 735 ICP-OES, Victoria, Australia). The grain was powdered and digested with a mixture of 3 mL $HNO_3$ (conc) and 1 mL of 30% $H_2O_2$ in a programmable microwave system. The following program was used to obtain a colorless solution: 90 °C, 6 min (750 W); 90 °C, 4 min (750 W); 180 °C, 8 min (1000 W); 180 °C, 15 min (1000 W); 0 °C, 20 min (0 W). Samples were diluted to a final volume of 10 mL with HPLC quality water [28].

## 2.7. Anatomical Evaluation

The anatomical characteristics of leaves and roots were assessed at three months after sowing. The samples were obtained from 10 plants from different pots and fixed in formalin–acetic acid–ethanol (1:1:18 v/v) solution. The cross-sections of samples were manually cut and double-stained with carmine and methyl green [29] and photographed using a light microscope (Olympus BX51) with an automatic camera at 100× and 400× magnifications. The following anatomical features were measured using Image Tools Version 3.0 software [26]: Leaf thickness (mm), diameter of vascular bundle (mm), width of xylem vessel area (mm), width of phloem vessel area (mm), width of sclerenchyma cell area (mm), length of bulliform cells of the leaves (mm), root diameter (mm), cortical parenchyma width (mm), vascular cylinder diameter (mm), width of xylem vessel area (mm), and width of Casparian strip of roots (mm).

The water quality standards for the safe and sustainable reuse of treated wastewater by the agriculture industry in different countries were described by Jeong et al. [30].

## 2.8. Statistical Analysis

The software SPSS (version 23) was used for analyzing the results. The data of $F_v/F_m$, $PI_{ABS}$, $P_N$, $C_i$, WUE, $T_r$, plant height, leaves area, and anatomical traits were obtained from means of 10 different plants from eight pots. The means of photosynthetic pigments, carbohydrates, glucose, and mineral elements are the average of four values for each treatment. Significant differences between data were tested by ANOVA. The means were compared by Duncan's test ($p \leq 0.05$).

## 3. Results

Large amounts of total suspended solids (TSS) were detected in the wastewater (Table 1). The analysis showed Ag, Cd, Co, Cu, Fe, Mn, Ni, Pb, and Zn, with Zn being the highest and Cd the lowest. The mineral elements Al, Ca, K, Mg, Na, P, S, and Si, were also detected with Na being highest and Si lowest. The Razi petrochemical complex also released high levels of $NH_4^+$, $NO3^-$, $PO_4^{3-}$, and $SO_4^{2-}$ into the wastewater (Table 1). The results of ICP analysis showed a significant increase in the levels of Al, Ca, Fe, Mg, Mn, Na, and S at all concentrations of wastewater in the leaves of both wheat cultivars, with the highest amounts in undiluted wastewater (Table 2). All levels of wastewater irrigation significantly reduced K, P, Si, and Zn in leaves of both cultivars, but the greatest reduction was with undiluted wastewater (Table 2). Ni accumulated in the leaves of both wheat cultivars, but not in control plants (Table 2). The Cu content in leaves of plants irrigated with undiluted wastewater significantly increased, while diluted wastewater had no effect on Cu in both wheat cultivars (Table 2).

The elements Ag, As, Ba, Be, Bi, Cd, Ce, Co, Cr, Li, Pb, Rb, Sb, Sc, Sn, Sr, Ti, U, and V were not detected in the leaves of either wastewater-treated plants or controls (Table 2).

**Table 2.** Elemental analysis of leaves of two wheat cultivars, Chamran and Behrang, irrigated with 0 (distilled water), 25%, 50%, and 100% (undiluted) wastewater from the Razi petrochemical complex. The same letters represent no significant difference at $p < 0.05$. DM, dry mass; ND, not detected.

| Element (mg/g DM) | Behrang Cultivar Wastewater Concentration | | | | Chamran Cultivar Wastewater Concentration | | | |
|---|---|---|---|---|---|---|---|---|
| | 0 | 25% | 50% | 100% | 0 | 25% | 50% | 100% |
| Ag | ND | ND | ND | ND | ND | ND | ND | ND |
| Al | 0.18 [e] | 0.22 [d] | 0.31 [c] | 0.69 [b] | 0.16 [e] | 0.24 [d] | 0.38 [c] | 0.78 [a] |
| As | ND | ND | ND | ND | ND | ND | ND | ND |
| Ba | 0.04 [b] | 0.04 [b] | 0.04 [b] | 0.12 [a] | 0.05 [b] | 0.06 [b] | 0.06 [b] | 0.13 [a] |
| Be | ND | ND | ND | ND | ND | ND | ND | ND |
| Bi | ND | ND | ND | ND | ND | ND | ND | ND |
| Ca | 18.8 [d] | 27.8 [c] | 40.4 [b] | 42.2 [b] | 19.4 [d] | 27.7 [c] | 45.6 [ab] | 49.1 [a] |
| Cd | ND | ND | ND | ND | ND | ND | ND | ND |
| Ce | ND | ND | ND | ND | ND | ND | ND | ND |
| Co | ND | ND | ND | ND | ND | ND | ND | ND |
| Cr | ND | ND | ND | ND | ND | ND | ND | ND |
| Cu | 0.01 [c] | 0.02 [bc] | 0.02 [bc] | 0.03 [b] | 0.01 [c] | 0.02 [bc] | 0.02 [bc] | 0.05 [a] |
| Fe | 0.24 [e] | 0.37 [d] | 0.47 [c] | 0.69 [b] | 0.24 [e] | 0.34 [d] | 0.46 [c] | 0.83 [a] |
| K | 43.4 [b] | 28.9 [d] | 14.7 [e] | 10.5 [f] | 55.6 [a] | 36.6 [c] | 25.3 [d] | 14.6 [e] |
| Li | ND | ND | ND | ND | ND | ND | ND | ND |
| Mg | 8.3 | 8.8 [cd] | 12 [c] | 20.1 [b] | 9.7 [c] | 11.8 [c] | 20 [b] | 25.2 [a] |
| Mn | 0.012 [cd] | 0.016 [c] | 0.022 [b] | 0.031 [a] | 0.013 [cd] | 0.019 [bc] | 0.024 [b] | 0.034 [a] |
| Mo | 0.01 [a] | 0.01 [a] | 0.01 [a] | 0.01 [a] | 0.01 [a] | 0.01 [a] | 0.01[a] | 0.01 [a] |
| Na | 1.8 [e] | 6.1 [d] | 9.8 [c] | 13.3 [b] | 2.04 [e] | 6.5 [d] | 14.7 [ab] | 16.5 [a] |
| Ni | ND | 0.02 [a] | 0.02 [a] | 0.02 [a] | ND | 0.02 [a] | 0.02 [a] | 0.03 [a] |
| P | 5.2 [a] | 4.6 [ab] | 3.9 [bc] | 3.3 [c] | 6.2 [a] | 5.5 [a] | 4.9 [ab] | 3.6 [bc] |
| Pb | ND | ND | ND | ND | ND | ND | ND | ND |
| Rb | ND | ND | ND | ND | ND | ND | ND | ND |
| S | 5.6 [de] | 7.6 [d] | 22.9 [bc] | 24.9 [b] | 6.8 [de] | 7.9 [d] | 25.4 [b] | 29.8 [a] |
| Sb | ND | ND | ND | ND | ND | ND | ND | ND |
| Sc | ND | ND | ND | ND | ND | ND | ND | ND |
| Si | 8.2 [a] | 5.8 [b] | 5.3 [b] | 4.5 [bc] | 7 [a] | 5.7 [b] | 4.6 [bc] | 4.1 [c] |
| Sn | ND | ND | ND | ND | ND | ND | ND | ND |
| Sr | ND | ND | ND | ND | ND | ND | ND | ND |
| Ti | ND | ND | ND | ND | ND | ND | ND | ND |
| U | ND | ND | ND | ND | ND | ND | ND | ND |
| V | ND | ND | ND | ND | ND | ND | ND | ND |
| Zn | 0.022 [a] | 0.007 [b] | 0.006 [b] | 0.002 [c] | 0.023 [a] | 0.01 | 0.009 [b] | 0.001 [c] |

The effect of wastewater on photosynthetic characteristics is illustrated in Figure 1. Wastewater irrigation at all levels significantly reduced both $F_v/F_m$ and $PI_{ABS}$ (Figure 1a,b). The highest and lowest reduction in fluorescence parameters was observed at 25% dilution and undiluted wastewater, respectively. With undiluted wastewater irrigation, $F_v/F_m/PI_{ABS}$ was reduced by about 59%/89% and 46%/82% in the Chamran and Behrang cultivars, respectively. The negative effect of wastewater on Chl fluorescence was higher in the Chamran cultivar than the Behrang. The response of $P_N$, WUE, $C_i$, and $T_r$ to wastewater irrigation was similar to the effects on fluorescence; they decreased significantly with increasing wastewater concentration, with undiluted wastewater causing the greatest reduction (Figure 1c–f). Undiluted wastewater irrigation reduced $P_N$ by about 90% in Chamran and 80% in Behrang, relative to controls. Undiluted wastewater reduced $C_i$ by 75% in Chamran and 61% in Behrang. The decline in WUE after undiluted wastewater irrigation was 88% in Chamran and 81%

in Behrang. The negative effect of undiluted wastewater on the $T_r$ parameter in Chamran, 66%, was greater than in Behrang at 58%.

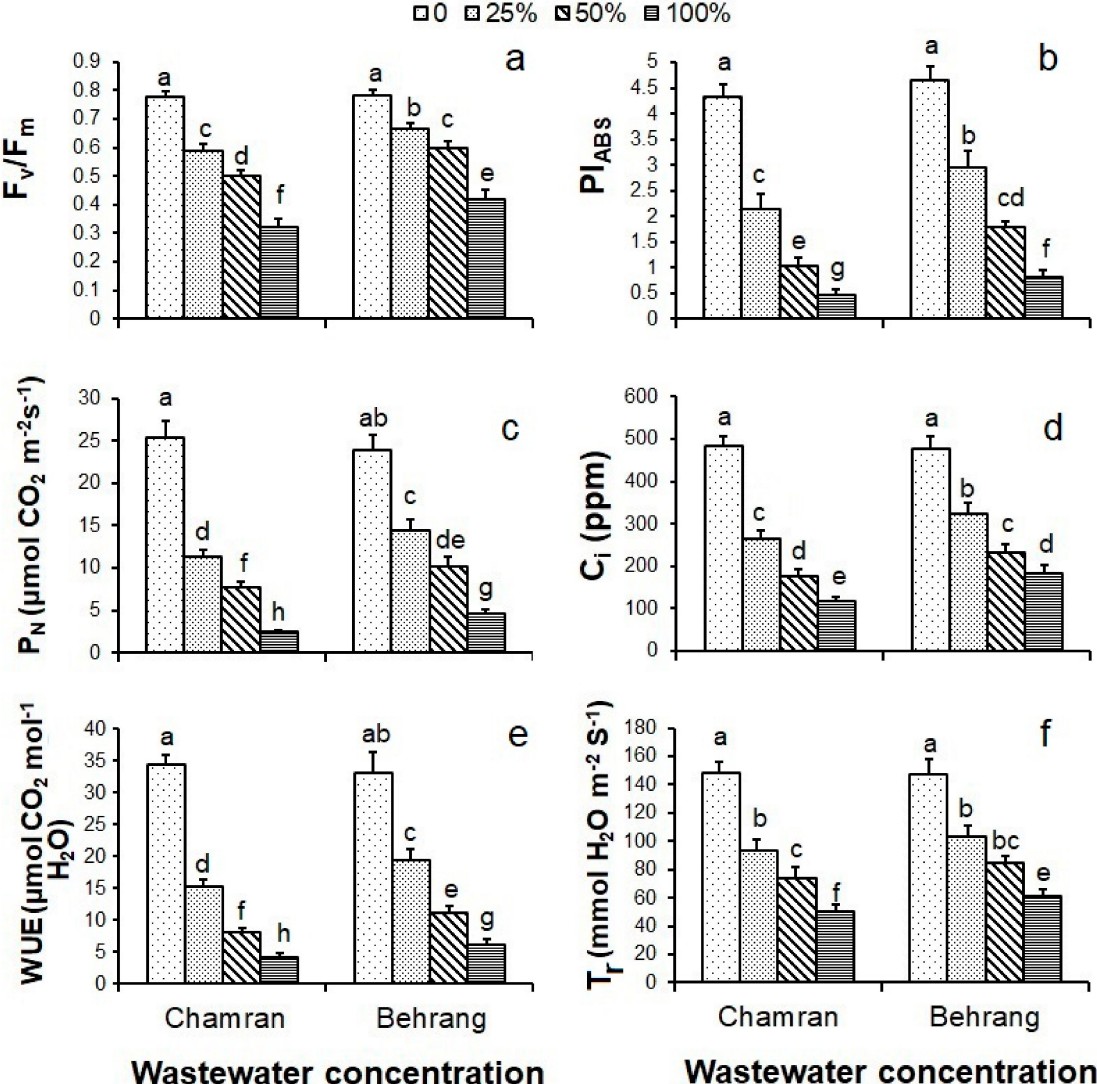

**Figure 1.** (**a**) Maximum quantum yield of photosystem II ($F_v/F_m$), (**b**) performance indices ($PI_{ABS}$), (**c**) net photosynthesis ($P_N$), (**d**) intercellular $CO_2$ ($C_i$), (**e**) water use efficiency (WUE), and (**f**) transpiration rate ($T_r$) of two wheat cultivars, Chamran and Behrang, irrigated with 0 (distilled water), 25%, 50%, and 100% (undiluted) wastewater from the Razi petrochemical complex. The same letters represent no significant difference at $p < 0.05$. The means are the averages of 10 plants. The error bars show standard deviation.

The negative effect of wastewater irrigation on photosynthesis was greater in the Chamran cultivar than the Behrang; photosynthetic pigments decreased in both cultivars (Figure 2). The amounts of Chl a, b, and total, and Car were significantly reduced with increasing wastewater concentration, and more so in Chamran than Behrang. The greatest decline in Chl a, b, and Car was with undiluted wastewater irrigation by about 95%, 91%, and 87% in Chamran and 85%, 83%, and 75% in Behrang (Figure 2a–d). The levels of photosynthetic pigments were almost the same in controls of both cultivars

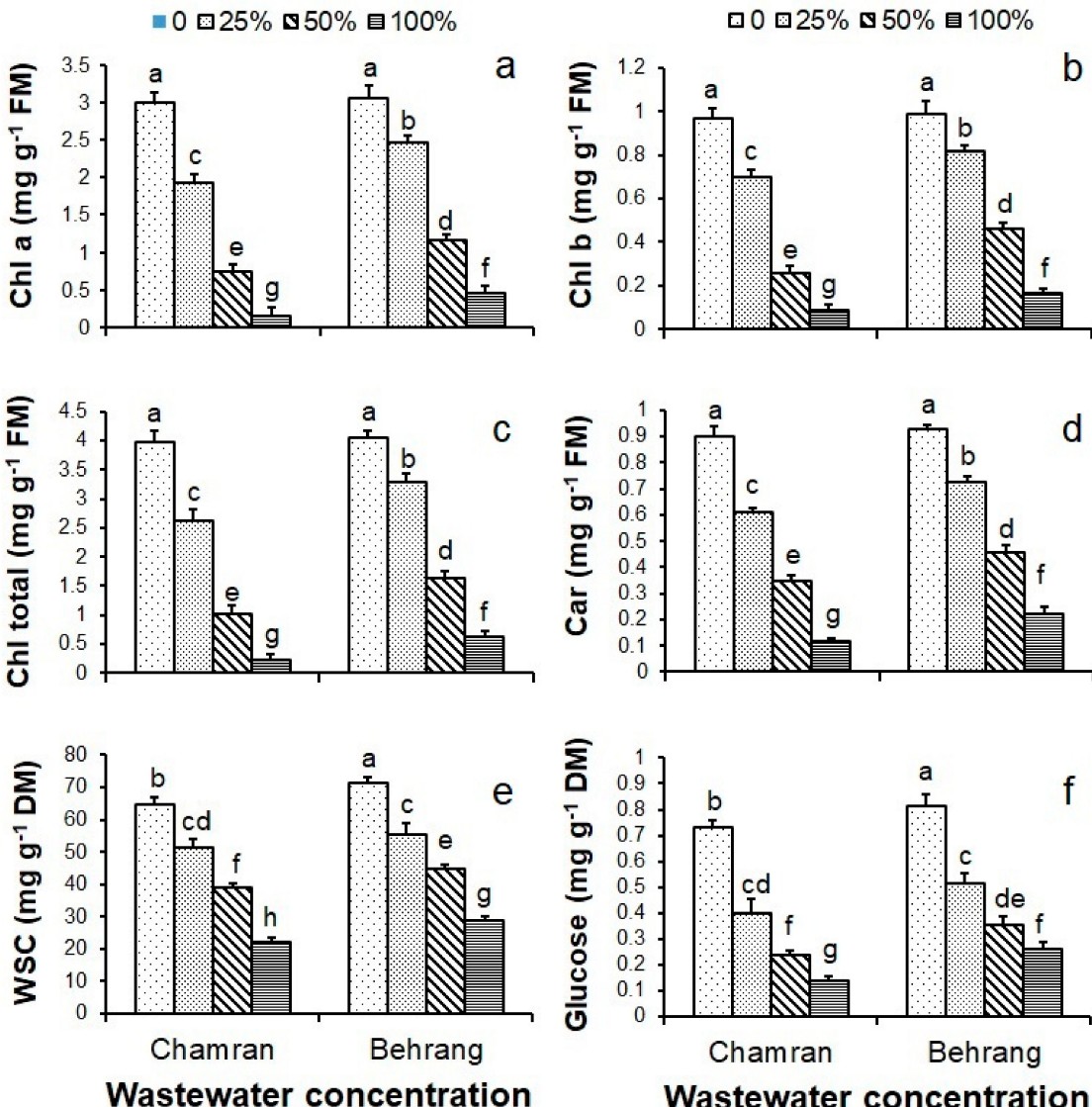

**Figure 2.** (**a**) Chlorophyll *a* (Chl *a*), (**b**) chlorophyll *b* (Chl *b*), (**c**) chlorophylls total, (**d**) carotenoids (car), (**e**) water-soluble carbohydrates (WSC), and (**f**) glucose of two wheat cultivars, Chamran and Behrang, irrigated with 0 (distilled water), 25%, 50%, and 100% (undiluted) wastewater from the Razi petrochemical complex. The same letters represent no significant difference at *p* < 0.05. The means are the averages of four values. The error bars show standard deviation.

The amounts of carbohydrates were also analyzed in wastewater-irrigated plants. The WSC data showed an adverse effect of wastewater treatment on carbohydrate biosynthesis in both cultivars (Figure 2). WSC and glucose significantly decreased with increasing wastewater concentration, and the greatest reduction occurred with undiluted wastewater (Figure 2e,f) 65% for WSC and 80% for glucose in Chamran, and 59% and 67% in Behrang.

The leaves are the main photosynthetic organs, and all concentrations of wastewater significantly reduced the leaf area in both cultivars, with the greatest decline being for undiluted wastewater—60% for Chamran and 58% for Behrang, relative to control (Figure 3a). Wastewater irrigation significantly decreased the height of both cultivars at all concentrations, with undiluted wastewater resulting in the greatest reduction in height (Figure 3b). The negative effect of undiluted wastewater irrigation on leaf area in the Chamran cultivar was greater than that in the Behrang cultivar by about 76% compared to 67%, respectively. The number of spikelets decreased with wastewater treatment, and none were seen in either cultivar with 100% wastewater. Irrigation with 25% and 50% wastewater significantly reduced

the spikelet number, with the most reduction at 50% (Figure 3c). Even though spikelets emerged with 50% wastewater treatment, no seeds formed in either of the cultivars. At 25%, the grain yield of both cultivars was significantly reduced by 95%. In control plants, the grain yield in Behrang was greater than in Chamran (Figure 3d).

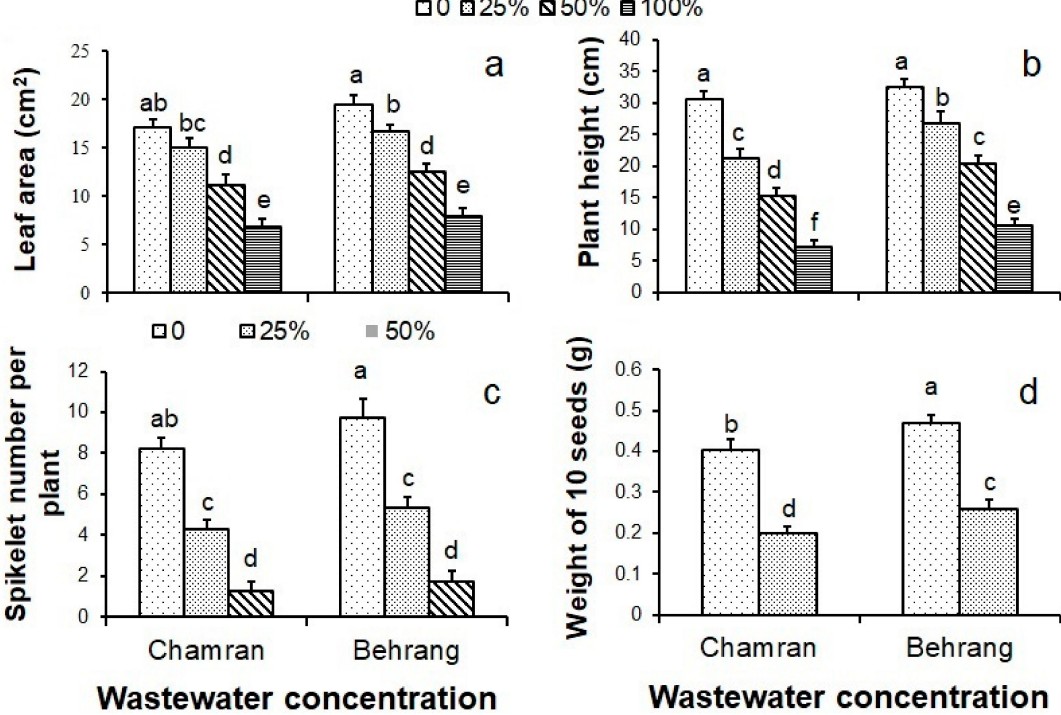

**Figure 3.** (**a**) leaf area, (**b**) plant height, (**c**) spikelet number per plant, and (**d**) weight of 10 seeds of two wheat cultivars, Chamran and Behrang, irrigated with 0 (distilled water), 25%, 50%, and 100% (undiluted) wastewater from the Razi petrochemical complex. The same letters represent no significant difference at $p < 0.05$ probability. The means are the averages of 10 plants. The error bars represent standard deviation.

The cross-sections of leaves (Figure 4) and roots (Figure 5) appeared different between controls and wastewater-irrigated cultivars. In the main vein area, the leaf thickness (Figure 6a), vascular bundle diameter (Figure 6b), width of xylem vessel area (Figure 6c), width of phloem vessel area (Figure 6d), and width of sclerenchyma cell area (Figure 6e) significantly decreased with increasing wastewater concentration in both cultivars. Wastewater induced larger bulliform cells in both cultivars (Figure 6f) compared to controls, with the largest bulliform cells occurring at 25% and 50%. Wastewater irrigation significantly decreased root diameter (Figure 7a), cortical parenchyma width (Figure 7b), vascular cylinder diameter (Figure 7c), width of xylem vessel area (Figure 7d), and Casparian strip width (Figure 7e) of both cultivars, with the greatest reduction at 100% wastewater.

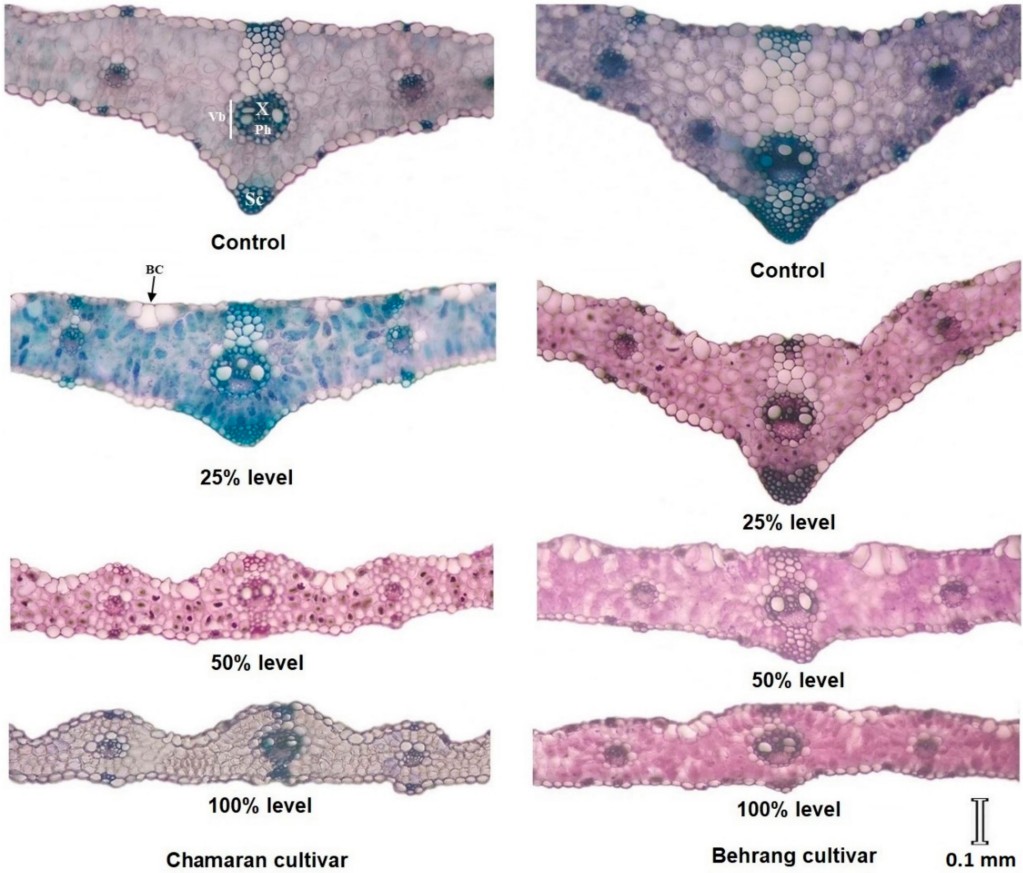

**Figure 4.** Micrographs of leaf cross sections of two wheat cultivars, Chamran and Behrang, irrigated with 0 (distilled water), 25%, 50%, and 100% (undiluted) wastewater from the Razi petrochemical complex. BC, bulliform cells; Ph, phloem vessels; Sc, sclerenchyma; Vb, vascular bundle; X, xylem vessels.

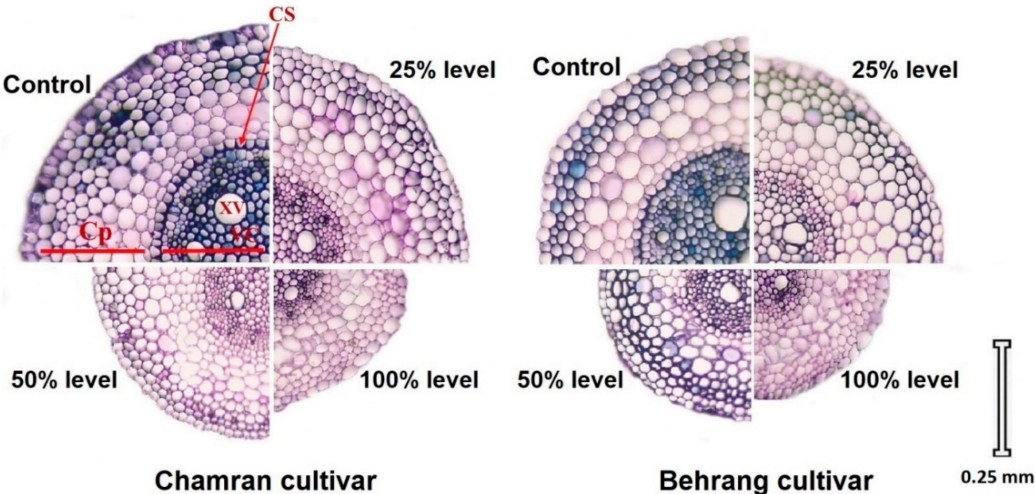

**Figure 5.** Micrographs of root cross sections of two wheat cultivars, Chamran and Behrang, irrigated with 0 (distilled water), 25%, 50%, and 100% (undiluted) wastewater from the Razi petrochemical complex. CP, cortical parenchyma; CS, Casparian strips; VC, vascular cylinder; XV, xylem vessels.

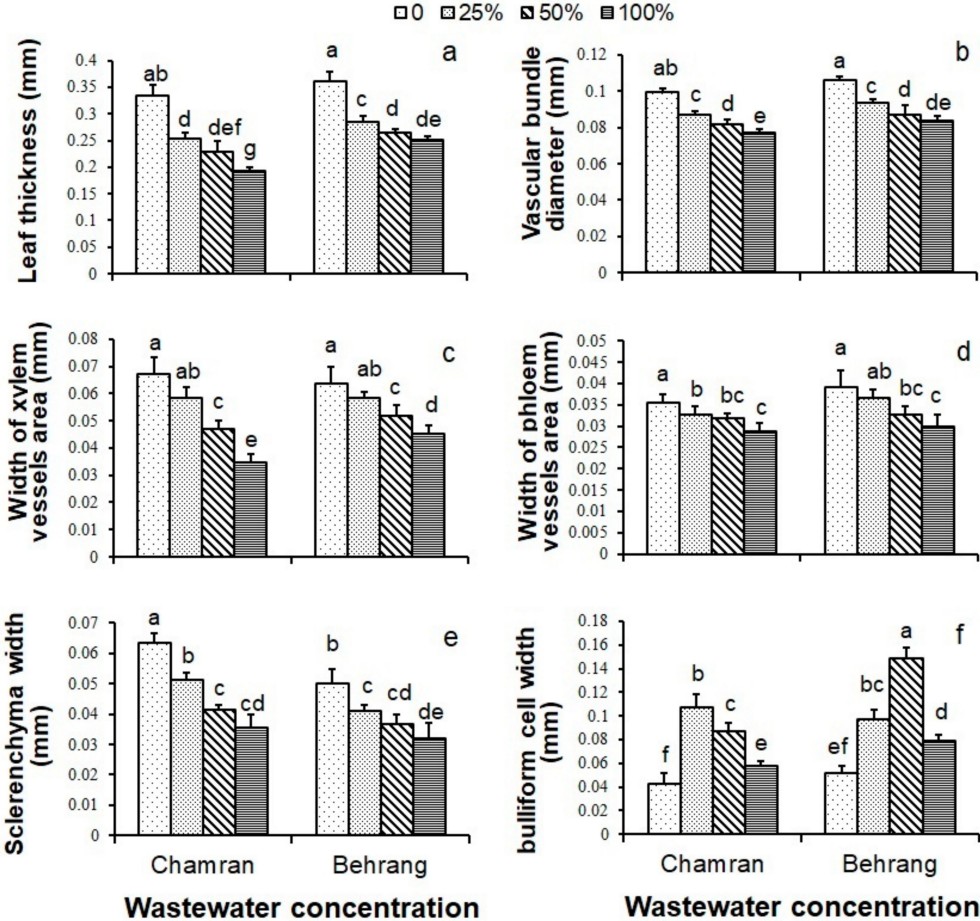

**Figure 6.** (**a**) leaf thickness in the midrib vein, (**b**) diameter of vascular bundle, (**c**) width of xylem vessel area, (**d**) width of phloem vessel area, (**e**) width of sclerenchyma cell area, and (**f**) width of bulliform cells of two wheat cultivars, Chamran and Behrang, irrigated with 0 (distilled water), 25%, 50%, and 100% (undiluted) wastewater from the Razi petrochemical complex. The same letters represent no significant difference at *p* < 0.05 probability. The means are the averages of 10 values. The error bars show standard deviation.

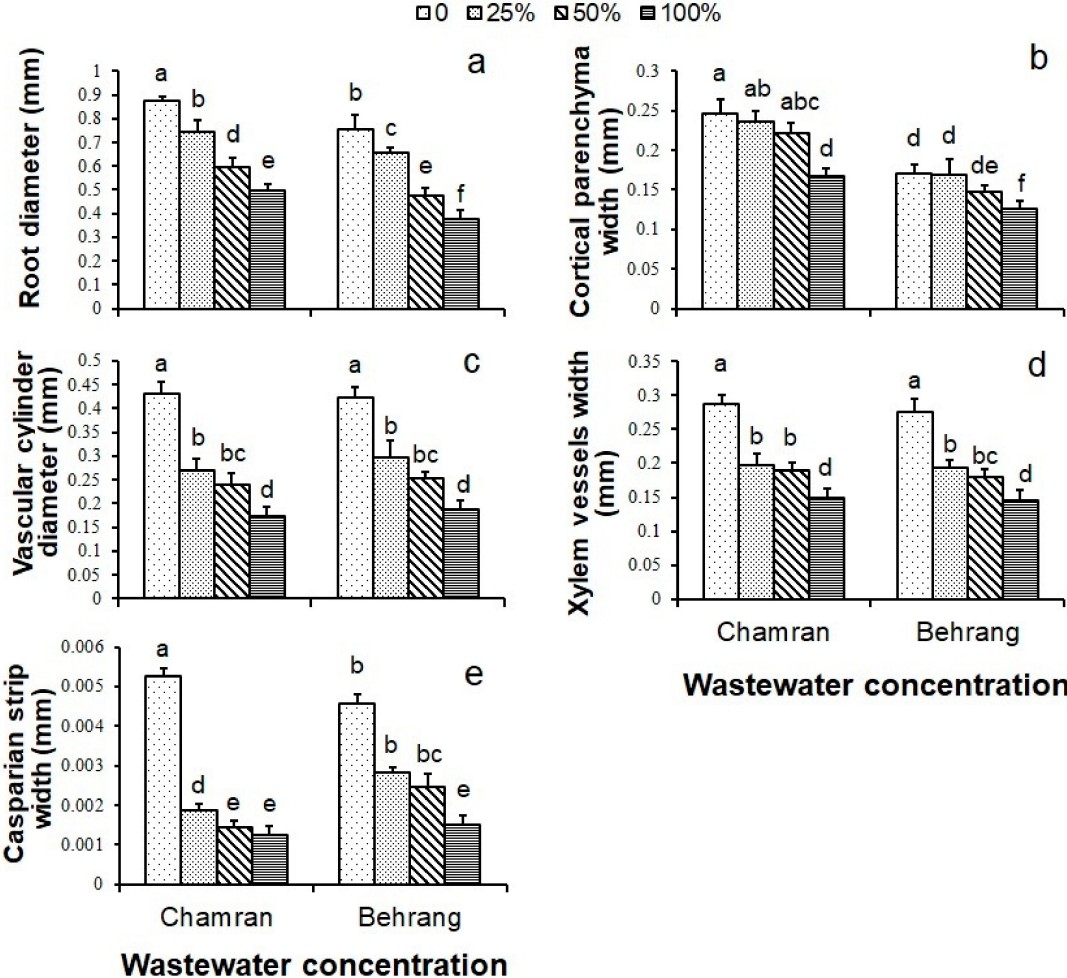

**Figure 7.** (**a**) Root diameter, (**b**) cortical parenchyma width, (**c**) vascular cylinder diameter, (**d**) width of xylem vessels, and (**e**) Casparian strip width of two wheat cultivars (Chamran and Behrang)-irrigated with 0 (distilled water), 25%, 50%, and 100% (undiluted) of wastewater obtained from Razi petrochemical complex. The same letters represent no significant difference at $p < 0.05$ probability. The means are the average of 10 values. The error bars show standard deviation.

## 4. Discussion

Water scarcity is one of the main problems for farmers, which tempts them to use wastewater for crop irrigation [1,31–34]. Tests of untreated wastewater from the Razi petrochemical complex showed high levels of TSS (325 mg $L^{-1}$). For comparison, the average amount of TSS in medium-quality water in Europe is < 35 mg $L^{-1}$, so the TSS level in Razi's untreated wastewater was about nine times greater than the allowable limit in Europe. The wastewater also contained high levels of Ca, K, Mg, Na, P, S, Si, $NH_4^+$, $NO^{3-}$, $PO_4^{3-}$, and $SO_4^{2-}$ The allowable values of chemical elements in wastewater for agricultural use varies based on the irrigation system, season, soil type, crop, and country standards. Based on FAO recommendations, the maximum values of trace elements in mg $L^{-1}$ are Al, 5.0; Cd, 0.01; CO, 0.05; Cu, 0.2; Fe, 1.0; Mn, 0.2; Ni, 0.2; Pb, 5.0; and Zn, 2.0 [30]. The amounts of Co, Cu, Fe, Mn, and Zn in Razi's wastewater were higher than the permissible values. Also, the Ag, Cd, Co, Cu, Fe, Mn, Ni, Pb, and Zn in the wastewater can accumulate in soil as a result of long-term wastewater irrigation of farmlands. The high concentrations of accumulated macronutrients (Ca, K, Mg, Na, P, S, and Si), micronutrients (Fe, Zn, Cu, and Mn), and heavy metals (Ag, Cd, Co, Ni, and Pb) from using untreated wastewater in agriculture can disrupt mineral nutrient uptake by plants, inhibit photosynthesis, reduce enzyme activity, interfere with physiological processes, damage cell membranes, limit biosynthesis of metabolites, and consequently decrease plant growth and yield [35–37]. Considering the high

concentrations of salts and metals measured in this study, especially toxic levels of Na and Ni in the wastewater and the sensitivity of wheat to salt stress, irrigation with wastewater from the Razi petrochemical complex was accompanied by a significant increase in amounts of Al, Ca, Fe, Mg, Mn, Na, and S and a significant decrease in K, P, Si, and Zn contents in the leaves of both cultivars, with the greatest reduction in Chamran. Cole et al. [37] reported that the combination of different nutrient elements did not always enhance growth and fruit yield of tomato plants and depended on their concentration. Excess macro- and micro-nutrients were toxic for wheat plants, reducing photosynthesis and biomass, and causing abnormal anatomical changes in both cultivars, but mostly in Chamran.

Wastewater irrigation significantly reduced the fluorescence characteristics with the greater effect again being on Chamran. With the high levels of salts and heavy metals present in the wastewater, the reduction in Chl-a fluorescence could be due to salt- or metal-induced stress. Oyiga et al. [38] reported that salt stress reduced Chl fluorescence characteristics in both tolerant and sensitive cultivars of wheat, but the sensitive cultivars were more severely affected. In our study, wastewater significantly reduced $P_N$ in both cultivars similarly to the effect on fluorescence. The reduction of $P_N$ could be the outcome of a lower quantum yield from PSII ($F_v/F_m$), diminished functionality of both PS I and II ($PI_{ABS}$), $CO_2$ gain ($C_i$), and loss of photosynthetic pigments, Chl-a, -b, and Car, induced by high levels of ions and metals (TSS). In accordance with these results, Oyiga [38] suggested that salt stress-induced reduction in photosynthesis resulted from the inhibition of PSII activity and reduction of chlorophylls and $CO_2$ assimilation in leaves due to the accumulation of toxic ions. Salt stress-induced stomatal closure resulted in an imbalance between light capturing and utilization, and a reduction in $CO_2$ entry and assimilation, which led to disturbances in the photosynthetic process in wheat [39,40]. WUE reduction, which results from stomatal closure and changes in plant water status, is a challenge for plants under salt stress [41]. High $Na^+$ levels interfere with $K^+$ uptake, which leads to disturbance in stomatal conductance [39]. This was seen in the correlation between increased $Na^+$ levels and reduced $K^+$ content in the leaves of both cultivars after wastewater irrigation, followed by problems in all photosynthetic processes. Stomatal conductance controls both water losses and $CO_2$ assimilation [42], so the reduction of both $C_i$ and $T_r$ in Chamran and Behrang can be attributed to abnormal stomatal function. Yang et al. [42] reported that increasing salt content reduced $P_N$ and $T_r$, which confirms these results. Nja et al. [43] reported that a low $T_r$ was accompanied by a low conductive rate in the narrow vessels due to salt stress. Similarly, the reduction in xylem vessels of roots and leaves of both cultivars was followed by a significant reduction in $T_r$, which can be explained by the high salinity of the wastewater. Moreover, the decline in WUE could be due to narrowing of xylem vessels that limits flow. The structure of xylem reflects the association between hydraulic conductance and the WUE of leaves in response to salinity [44]. One of the main symptoms of salt stress is chlorosis in plants [40]. In accordance with the results of this study, the reduction in photosynthetic pigments was also reported by other researchers [16,38–40]. Even though Zn is a necessary cofactor for enzymes participating in carbohydrate metabolism and other physiological processes [45], it is harmful at high levels because Zn can displace the Mg cofactor in chlorophyll, thus disrupting photosynthesis and limiting growth [46]. The high Zn levels in wastewater could be one of the main reasons for the decline in chlorophyll. In wastewater-irrigated plants, the reduced photosynthetic capacity from high levels of $Na^+$ [38] caused a reduction in WSC and glucose that was highest in the Chamran cultivar. The P-containing molecules, ATP and NADP, are essential for utilizing the energy of photosynthesis to make sugars [47], so the reduction of P in the leaves of wastewater-irrigated plants could be another reason for the reduction in carbohydrates and consequently in plant growth. Thus, the loss of carbohydrates in response to wastewater could be attributed to nutrient imbalance and impaired photosynthesis. Accumulation of $Na^+$ in the leaves of wastewater-irrigated plants can explain the reduced leaf area because high levels of $Na^+$ reduce the supply of carbohydrates to young leaves [38].

Depression of plant growth can be due to salt-induced osmotic stress that reduces water absorption by roots and leaf water status in wheat [39]. Wastewater irrigation caused anatomical changes in the leaves and roots of Chamran and Behrang cultivars. The reduction in xylem vessels and in development

of the Casparian strip was another result of wastewater treatment, with the highest reduction at 100%. Hwang and Chen [48] reported an increase in the Casparian strip as a general characteristic of salt-tolerant plants growing in saline habitats, which is the opposite of our findings. Under salt stress, the Casparian strip acts as a barrier to prevent salt influx into the xylem through the apoplast pathway [49]. The reduction in the radial width of the Casparian strip in wastewater-irrigated wheat cultivars would lead to an increase in the influx of salts and heavy metals into the xylem vessels and their transfer to the leaves, which is evidenced by the high levels of Na, Al, Ca, Fe, Mg, Mn, S, Ni, and Cu in leaves of these plants. The diameter of the root cross sections in the control plants was larger than in wastewater-irrigated plants because of the greater cortical thickness and stele diameter in controls. Similar results were observed in *Kandelia candel* in response to high levels of salt [48]. In the early stages of wheat growth, the salt stress-induced reduction in size due to a shorter length and a narrower thickness was accompanied by changes in leaf physiological functions [50]. Similar to the roots, there was a decrease in thickness of leaves in the midrib veins along with a reduction in mesophyll width and vascular bundle diameter in response to wastewater irrigation. Hu et al. [50] reported that salt stress only decreased the cross-sectional area of leaves in the midrib and large veins, which agrees with results here. The reduction of xylem vessel area in the leaves' midrib veins can lead to the reduction of water flow to the leaves as an active sink for water. The reduction of water flow to the leaves via midrib and large veins under saline condition leads to disturbances in the transport of mineral nutrients along with loading and transport of carbohydrates [50], which would explain the observed reduction in mineral nutrients in leaves and seed yield. The xylem tissues are less permeable in the presence of NaCl, which reduces water flow into the leaves and water-use efficiency [44]. The reduction in phloem area in the main vein with increasing wastewater percentage was followed by reduction in carbohydrate transport from source (leaf) to sink (seed) and diminished seed yield. An increase in phloem area had been reported in salt-tolerant plants [51], so the opposite results of the present study would suggest that the Chamran and Behrang cultivars are intolerant to high levels of salts in wastewater. The abundance of sclerenchyma cells in the leaves of salt-tolerant plants are important for rigidity and for reducing water loss under high salinity [51]. The fewer sclerenchyma cells in wastewater-irrigated plants may indicate that these wheat cultivars were salt-sensitive and that the high salt levels induced less sclerification in them. One other interesting anatomical change in response to wastewater irrigation was a significant increase in bulliform cells in both cultivars, which was more prominent in Behrang than Chamran. The enlargement of bulliform cells causes rolling of leaves to limit water loss under during salt stress [51], hence the extensive bulliform cells observed in the wastewater-irrigated cultivars could be an important defensive strategy against drought stress induced by the high salt content in the wastewater.

In accordance with the observed reduction in seed yield of Chamran and Behrang cultivars in response to wastewater irrigation, Sun et al. [39] reported a large reduction in grain yield of two wheat cultivars in response to salt stress, with a higher reduction in the salt-sensitive cultivar than the salt-tolerant one. The data of the present study showed that grain filling decreased in wastewater-irrigated plants, which could be explained by impaired photosynthesis resulting in reduced carbohydrate biosynthesis and abnormalities in phloem vessels of leaves that inhibited translocation to the sink organ (seeds). Overall, the plant height, leaf area, and grain yield were significantly depressed in response to increasing wastewater concentration in both wheat cultivars, with a relatively higher reduction in Chamran than Behrang.

## 5. Conclusions

Wheat irrigation with untreated wastewater from the Razi petrochemical complex decreased the photosynthetic characteristics, nutrient concentrations, and carbohydrates, which resulted in reduction in growth. The lower carbohydrate accumulation in wastewater-irrigated plants suggests that critical damage to the photosystems was responsible for the reduction in grain yield. The anatomical study suggests that the formation of larger bulliform cells in the leaves of wastewater-irrigated plants

represents the induction of a defense system against water loss. The degree of decline in the analyzed parameters was related to genotype, which was more severely affected in the Chamran cultivar than in the Behrang. Accordingly, the reusing of untreated wastewater with high TSS is not suggested for crop irrigation. These results supported the idea that wastewater dilution will not necessarily reduce its toxicity and that the use of wastewater without any processing is dangerous for the environment and for human health. The data presented here also confirmed the advantages of using the rapid tool of Chl fluorescence to monitor the adverse effects of wastewater irrigation on plants at the early stage of crop growth.

**Author Contributions:** Conceptualization, S.H.; methodology, S.H., F.N., M.R.; software, S.H, M.S.; validation, S.H., M.B.; formal analysis, S.H.; investigation, S.H.; resources, S.H.; data curation, S.H.; writing—original draft preparation, S.H.; writing—review and editing, S.H., M.S., M.B.; visualization, S.H.; supervision, S.H., M.B.; project administration, S.H., M.B.; funding acquisition, S.H., M.B, M.S.. All authors have read and agreed to the published version of the manuscript.

**Funding:** This research was founding by APVV Agency, project No. APVV-18-0465.

**Conflicts of Interest:** The authors declare no conflict of interest.

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
