# Peer review of "Effect of Wastewater Irrigation on Photosynthesis, Growth, and Anatomical Features of Two Wheat Cultivars (Triticum aestivum L.)"

_water, doi:10.3390/w12020607_

Round 1

Reviewer 1 Report

General Comments

The present paper deals with the reuse of wastewater from a petrochemical complex for agricultural irrigation purposes. The authors study the response of two wheat cultivars to water pollution. The authors justify their study with the concern of alleviating water scarcity through the reuse of industrial wastewater for irrigation.

However, due to the followed approach, some confusion on the practice of using wastewater for irrigation purposes without adequate treatment might arise. Such a practice is not recommended, and it is prohibited in most countries. What is recommended, to avoid water scarcity, is the reuse of treated wastewater (reclaimed water), not the reuse of wastewater directly. Furthermore, to avoid the contamination of crops, soil and public health issues, the treated wastewater needs to meet minimum requirements in order to be used for agricultural irrigation. The authors should clarify these aspects in the introduction, referring to the legislation in force in the countries involved in the study and in the EU. In addition, they must also provide data on the reclaimed water quality requirements for agricultural irrigation.

The authors studied various dilutions of wastewater. However, it should be noted that dilution is not a form of wastewater treatment and will only contribute to increasing water scarcity.

Finally, the manuscript is well structured, but is poorly written and has significant grammatical errors, which makes it difficult to understand. I recommend that the text should be carefully revised.

Detailed Comments

Abstract

Line 19-20, Please add total suspended solids (TSS).

Introduction

I recommend completing the introduction with the comments described above in the general comments.

Materials and methods

Please complete table 1 with the water quality standards for reclaimed water, using for example the EU standards.

Results

Compare the results of wastewater analysis with the water quality standards.

Table 2 – define “DM” and “ND”

Caption of figure 2 – Please check the correspondence of the letters with the graphics, because some of them are not correct.

Line 261, “Fig.3” or Fig. 2 ?

Line 263 “(Fig. 3 a, b)” or (Fig. 2 e, f) ?

Caption of figure 3 - Check the correspondence of the letters with the graphics, because some of them are not correct.

Line 275, “(Fig. 4a) or (Fig. 3a)?

Line 278, “(Fig. 4b)” or (Fig.3b)?

Discussion

Line 323, I suggest writing “Razi petrochemical complex wastewater” instead of “wastewater”

Line 324-325, Add a reference for the allowable value of TSS and compare the other elements and pollutants with the allowable Quality Standards too.

I recommend adding to the discussion the above general comments related to the use of wastewater for agricultural irrigation purposes.

Line 337, Please write “Cole et al.” instead of “JC Cole, MW Smith, CJ Penn, BS Cheary and KJ Conaghan”. Check and correct similar errors throughout the manuscript, whenever there are more than two authors.

Conclusions

Reinforce that untreated wastewater should not be used to irrigate agricultural crops. That practice can be harmful to crops, contaminate soils and may cause public health problems. To be reused, wastewater must undergo primary, secondary and advanced treatments. Dilution of wastewater is not an adequate form of treatment and only contributes to water scarcity.

Author Response

Dear Reviewer 1:

Thank you very much for reviewing our manuscript and for your insightful comments. We have revised the manuscript according to your suggestions.

Response: We agree with you that wastewater dilution is not a wastewater treatment and is dangerous for the environment and nature but unfortunately, it is done in some areas.

In our research, we pointed out the inhibition of photochemical and non-photochemical processes of photosynthesis as well as the efficiency of water utilization by using untreated wastewater for irrigation. Such treatment causes stress in the plants and is reflected in sensitive and exact physiological parameters such as Fv / Fm and PI, which are used as standard criteria to evaluate the effect of stress on plants. Our paper demonstrates that irrigation with wastewater can be counterproductive and can have negative effects on plants, even in water use efficiency.

To support our conclusions, we added some new references to the revised manuscript.

Our experiment were designed to prove that wastewater dilution could not reduce its harmful effects and we followed your suggestion and elaborated on this critical point in the revised manuscript.

The English was corrected and improved by a professional English editing service.

Detailed Comments

Abstract

Line 19-20, Please add total suspended solids (TSS).

Response: Thank you very much for this comment. We have added TSS.

Introduction

I recommend completing the introduction with the comments described above in the general comments.

Response: Thanks for your positive and constructive comments. We have carefully improved the revised manuscript for details.

Materials and methods

Please complete table 1 with the water quality standards for reclaimed water, using for example the EU standards.

Response: Thanks a lot for this comment. We agree with you, but different European countries have different standards and there is a manuscript by Hanseok Jeong, Hakkwan Kim, and Taeil Jang entitled “IrrigationWater Quality Standards for Indirect Wastewater Reuse in Agriculture: A Contribution toward Sustainable Wastewater Reuse in South Korea” published in Water which explained about standards in different countries and FAO, so we referenced it for readers to refer to for more information.

Results

Compare the results of wastewater analysis with the water quality standards.

Response: We have compared it with FAO standards based on the 2016 Jeong et al.paper.

Table 2 – define “DM” and “ND”

Response: Thanks a lot for this comment. It is done.

Caption of figure 2 – Please check the correspondence of the letters with the graphics, because some of them are not correct.

Line 261, “Fig.3” or Fig. 2 ?

Line 263 “(Fig. 3 a, b)” or (Fig. 2 e, f) ?

Caption of figure 3 - Check the correspondence of the letters with the graphics, because some of them are not correct.

Line 275, “(Fig. 4a) or (Fig. 3a)?

Line 278, “(Fig. 4b)” or (Fig.3b)?

Discussion

Line 323, I suggest writing “Razi petrochemical complex wastewater” instead of “wastewater”

Line 324-325, Add a reference for the allowable value of TSS and compare the other elements and pollutants with the allowable Quality Standards too.

I recommend adding to the discussion the above general comments related to the use of wastewater for agricultural irrigation purposes.

Line 337, Please write “Cole et al.” instead of “JC Cole, MW Smith, CJ Penn, BS Cheary and KJ Conaghan”. Check and correct similar errors throughout the manuscript, whenever there are more than two authors.

Response: Thank you very much for your attention and for indicating careless mistakes. The mentioned writing problems have been corrected.

Conclusions

Reinforce that untreated wastewater should not be used to irrigate agricultural crops. That practice can be harmful to crops, contaminate soils and may cause public health problems. To be reused, wastewater must undergo primary, secondary and advanced treatments. Dilution of wastewater is not an adequate form of treatment and only contributes to water scarcity.

Response: Yes, thank you for your comment. We have restated the conclusion in the revised manuscript to make it more accurate and complete.

Your comments helped us a lot to improve our manuscript and we appreciated your careful attention. We hope that the edited manuscript is now suitable for publication in Water.

 Sincerely yours,

Authors

Reviewer 2 Report

The reuse of wastewater for irrigation in agriculture is a method that is widely used in recent years. There have been many studies which gave detailed information about assessment of soil and water pollution due to waste water application. This study is trying to assess the effect of wastewater on physiological, morphological, anatomical and yield of both wheat cultivars, reusing wastewater with high levels of salts. The authors conclude that irrigation with wastewater cannot be suggested for the wheat crop. However, the study do not indicate if there available data for inorganic nitrogen, other soluble ion or salinity. And probably, to be able to present convincing values in this regard, the study should be repeated every year for at least two or three years.

I suggest to include a map shown the área in study. I also suggest to delette in table 2 the elements with ND; please explain it in the text. Finally I suggest to reduce the discussion.

Author Response

Dear Reviewer 2:

Comments and Suggestions for Authors

The reuse of wastewater for irrigation in agriculture is a method that is widely used in recent years. There have been many studies, which gave detailed information about assessment of soil and water pollution due to wastewater application. This study is trying to assess the effect of wastewater on physiological, morphological, anatomical and yield of both wheat cultivars, reusing wastewater with high levels of salts. The authors conclude that irrigation with wastewater cannot be suggested for the wheat crop. However, the study does not indicate if their available data for inorganic nitrogen, other soluble ion or salinity. And probably, to be able to present convincing values in this regard, the study should be repeated every year for at least two or three years. I suggest to include a map shown the area in study. I also suggest to delete in table 2 the elements with ND; please explain it in the text. Finally, I suggest to reduce the discussion.

Response: Thank you very much for your attention and comments. We are happy for the chance to eliminate weaknesses in our manuscript based on your suggestions. There were some points that we had not appreciated so we tried to follow all of your comments and they were useful in improving our manuscript.

Response: You are completely right and we suggested not to irrigate plants with wastewater; but, actually what we said was to not use untreated wastewater. In the revised manuscript, we explicitly state that crops should not be irrigated with untreated wastewater. Thank you for your critical comments about our manuscript and about inorganic nitrogen. It is a good suggestion for a future study; but, the main aim in the present study was to evaluate physiological, morphological and anatomical parameters, which is why we did not measure N.

As stated in the manuscript, the plant culture and treatment experiments were independently repeated three times. As whole experiments were done under controlled conditions and they were pot experiments, they were done continually. Each experiment included a set of eight pots for each treatment, with six plants per pot to provide enough plant mass for physiological, morphological and anatomical studies.

Regarding the location of the Razi petrochemical complex, we are sorry to state that we cannot specify the geographical site by law.

You are completely right and the data of ND elements in the table makes it rather busy, but as readers would rather see the data easily in a table than reading the text, we placed it in table 2.

Regarding the Discussion, we have revised the section to make it more objective and focused.

Again, thanks for your great comments. We hope that the revised manuscript now meets your standards for publication in Water

Sincerely yours,

Authors

Round 2

Reviewer 1 Report

The authors made all the suggested changes and significantly improved the paper.